# Provably Secure ECC-Based Anonymous Authentication and Key Agreement for IoT

**Shunfang Hu** *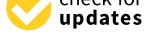**, Shaoping Jiang, Qing Miao, Fan Yang, Weihong Zhou and Peng Duan**

School of Mathematics and Computer Science, Yunnan Minzu University, Kunming 650504, China; shaopingjiang@ynni.edu.cn (S.J.); 041087@ynni.edu.cn (Q.M.); 040879@ynni.edu.cn (F.Y.); 040300@ynni.edu.cn (W.Z.); 040268@ynni.edu.cn (P.D.)
* Correspondence: hsf@ymu.edu.cn

**Abstract:** With the rise of the Internet of Things (IoT), maintaining data confidentiality and protecting user privacy have become increasingly challenging. End devices in the IoT are often deployed in unattended environments and connected to open networks, making them vulnerable to physical tampering and other security attacks. Different authentication key agreement (AKA) schemes have been used in practice; several of them do not cover the necessary security features or are incompatible with resource-constrained end devices. Their security proofs have been performed under the Random-Oracle model. We present an AKA protocol for end devices and servers. The proposal leverages the ECC-based key exchange mechanism and one-way hash function-based message authentication method to achieve mutual authentication, user anonymity, and forward security. A formal security proof of the proposed scheme is performed under the standard model and the eCK model with the elliptic curve encryption computational assumptions, and formal verification is performed with ProVerif. According to the performance comparison, it is revealed that the proposed scheme offers user anonymity, perfect forward security, and mutual authentication, and resists typical attacks such as ephemeral secret leakage attacks, impersonation attacks, man-in-the-middle attacks, and key compromise impersonation attacks. Moreover, the proposed scheme has the lowest computational and communication overhead compared to existing schemes.

**Keywords:** authentication and key agreement; anonymity; Internet of Things; standard model; elliptic curve cryptography

## 1. Introduction

Thanks to advances in chipset production and embedding technologies, sensors and actuators (referred to as end devices) are pervasive in the Internet of Things (IoT), being integrated into intelligent agriculture, smart grid (SG), telemedicine, smart home, intelligent manufacturing, and many other fields to collect and disseminate the data [1]. According to the latest estimates, there will be 83 billion IoT connections by 2024 [2]. In IoT applications, the collected and transmitted data are sensitive. Privacy is another crucial issue, especially regarding user data such as consumption habits, location, and communication activities [3,4]. To ensure security, authentication key agreement (AKA) schemes for IoT applications have been widely used, which offer mutual authentication and privacy protection and ensure confidentiality, integrity, and non-repudiation of data transmissions based on the negotiated session keys [5]. End devices are often linked to open networks and deployed in unattended environments with limited computation, communication, and storage capabilities. As a result, mutual authentication and key agreement between end devices and servers to sustain efficiency is a critical challenge.

### 1.1. Related Work

Over the last few years, numerous AKA solutions have been developed for IoT applications. The symmetric cryptography-based AKA protocols [6–9] have the advantages of low

computational complexity and high efficiency. On the other hand, such schemes necessitate sharing key parameters between end devices beforehand or each device transferring its key to the server. It is unrealistic for numerous end devices and significantly burdens the servers. Physical Unclonable Function (PUF) is a promising lightweight hardware security primitive adopted by many IoT AKA protocols [10–12]. Each participant in these schemes should record one or more Challenge–Response Pairs (CRPs) of its PUF with the registration server beforehand. When a registered device, Alice, wants to communicate with another registered device, Bob, it can only do so with the assistance of the server, which results in a lack of flexibility and efficiency. In contrast, the asymmetric cryptography-based AKA schemes requiring fewer restrictions have attracted increasing attention [13]. Elliptic Curve Cryptography (ECC) provides smaller key sizes than other asymmetric algorithms with the same security [14,15], which makes it introduced in IoT AKA protocols.

Numerous IoT AKA protocols based on ECC have been developed. In 2015, a bilinear pairing-based AKA protocol for wireless body area networks (WBAN) was put forward by Wang et al. [16], which requires a high computational overhead. They claimed that their scheme achieves absolute anonymity, perfect forward security (PFS), and overcomes the weaknesses of previous schemes. After analysis, it was found that the session key could be captured after temporary session information disclosure. In addition, Wu et al. [17] pointed out that the protocol is incapable of withstanding impersonation (IM) attacks. And then, they proposed an enhanced version for WBANs. However, the enhanced scheme also uses bilinear pairing and suffers from ephemeral secret leakage (ESL) attacks. Seo et al. [18] introduced an AKA scheme for dynamic WSNs. Later, Saeed et al. [19] pointed out that the scheme [18] could not provide PFS; then, they proposed a scheme for establishing an authenticated key between WSNs and cloud servers, whereas the proposal [19] is also not resistant to ESL attacks and cannot provide user anonymity. In 2020, an AKA scheme for IoT was introduced by Fang et al. [20]. In this scheme, heterogeneous-type IoT smart devices are deployed based on a trust model. Their solution requires higher computational and communication costs and is susceptible to ESL attacks [21]. In the same year, Dariush et al. [22] introduced an AKA protocol for SG that offers solutions to some of the previously mentioned problems, such as ESL attacks and private key leakage attacks. Unfortunately, in [22], the trusted authority (TA) is able to masquerade as a smart meter to agree on session keys with the server provider [23]. Moreover, the scheme needs more computational and communication costs for the bilinear pairing computation.

Recently, Srinivas et al. [24] designed an anonymous AKA protocol with Schnorr's signature. Later, Baruah et al. [23] demonstrated that the scheme [24] is prone to man-in-the-middle (MIM) attacks and IM attacks. Cryptanalysis identifies that the protocol [24] is also vulnerable to key escrow problems and ESL attacks. Yang et al. [25] stated that Shen et al.'s scheme [26] suffers from MIM attacks and key compromise impersonation (KCI) attacks and is incapable of providing PFS, and then introduced an enhanced cloud-based scheme. Unfortunately, the enhanced scheme has key escrow problems and is incapable of providing user anonymity. Chaudhry et al. [27] presented an AKA scheme for SG using ECC and symmetric encryption. Unfortunately, this scheme [27] has key escrow problems and suffers from MIM attacks. Hajian et al. [28] examined the deficiencies of four existing AKA schemes and then proposed an improved device-to-device AKA scheme in the IoT. But the improved scheme suffers from MIM attacks and KCI attacks and is incapable of affording PFS. In 2023, Chen et al. [29] presented an AKA scheme for industrial control systems. However, the solution requires high computation and communication costs, suffers from ESL attacks, and cannot afford PFS.

### 1.2. Related Adversary Model

In 1993, Bellare and Rogaway [30] put forward an adversary model for the AKA scheme, the BR model, which formalized the attacker's known-key attacks and IM attacks. Later, the BR model was modified by Blake-Wilson et al. [31] by introducing long-term private key corruption attacks. In 2001, Canetti and Krawczyk [32] proposed the CK model,

which covers attacks on ephemeral private keys and intermediate result leakage. All these adversary models attempt to cover the essential safety and performance attributes required. In 2007, LaMacchia et al. [33,34] introduced a somewhat stronger adversary model, the extended CK model (eCK model), which incorporates weak PFS and KCI attacks.

### 1.3. Random-Oracle Model and Standard Model

Provable security theory, which employs formal language to describe the security of cryptographic protocols, has played a critical role in designing and analyzing AKA protocols. Most early cryptographic schemes for provable security were inefficient. Practically oriented provably secure cryptographic schemes were proposed only after the famous Random-Oracle model was introduced by Bellare and Rogaway [35]. In the Random-Oracle model, the hash function is treated as a completely randomized machine called $R$, and the adversary has no access to its information. A random oracle is a theoretical model that takes deterministic inputs and produces random outputs. Finding a genuinely random function to replace the random oracle $R$ in the Random-Oracle model is impossible. Many scholars have suggested avoiding using hash functions as random oracles in favor of designing cryptographic protocols directly under realistic conditions [36–38]. This approach, called the standard model, avoids using idealized models such as hash functions. In general, cryptographic schemes that are provably secure under the standard model can provide more robust security than those that are provably secure under the randomized predicate model.

### 1.4. Motivation and Contributions

To summarize, previous ECC-based AKA schemes suffer from more or less vulnerabilities, i.e., failure to provide user anonymity [19,25], PFS [18,24,28,29], and vulnerability to specific attacks [16–20,22,24,25,27–29]. Next, high computational and communication costs eliminate the suitability of some solutions for resource-limited IoT [10,16,17,20,22,29]. Their security proofs are performed in the Random-Oracle model model [22,24]. It is attractive to design an efficient AKA scheme for IoT and provide security proof under the standard model and eCK model.

We propose an improvement over the scheme of Srinivas et al. [24] with the ECC-based message exchange mechanism and the one-way hash function message authentication technique. During registration, the TA only possesses part of the entity's private key, solving the key escrow issues. In addition, the proposals provide PFS and can resist ESL attacks since session keys are generated using both long-term and ephemeral credits. The protocol encrypts entity identities dynamically with random numbers and transmits them anonymously from session to session.

The paper's contributions can be summarized as follows:

(1) As an example, the cryptanalysis of the protocol scheme of Srinivas et al. [24] for the previous scheme reveals security issues and vulnerabilities.

(2) A secure-enhanced AKA protocol for IoT is presented. Its security is formally proved under the standard model and the eCK model with the elliptic curve encryption computational assumptions and verified with ProVerif.

(3) The proposed protocol has better security features with lower communication and computational overheads than existing schemes.

### 1.5. Roadmap

The paper is structured as follows: Section 2 reviews the network model and the basics of elliptic curve encryption. In Section 3, we analyze a related AKA scheme. We then describe an improved ECC-based AKA protocol in Section 4. Section 5 provides a formal proof, descriptive security analysis, and validation with ProVerif of the proposed scheme security. In Section 6, we present a performance comparison with related schemes. Finally, we conclude the paper in Section 7.

## 2. Preliminaries

The following preliminaries and symbols are used to explain and analyze the schemes.

### 2.1. Network Model

A typical IoT application is shown in Figure 1. It mainly involves three main components: end devices, routers, and servers. The end devices may be sensors, actuators, cell phones, etc. Routers include gateway nodes, base stations, and routers for relaying and passing messages. In addition, servers are in charge of managing devices and assigning security parameters.

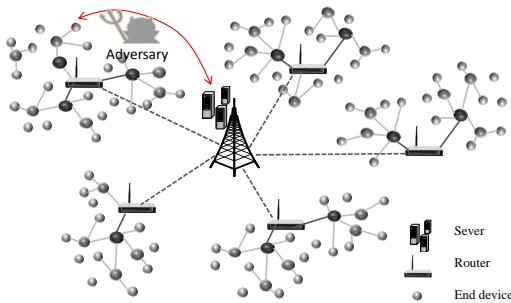

**Figure 1.** Network model.

An IoT system consists of many low-power, resource-limited end devices placed in unattended or open environments and typically connected to open networks. Through these terminal devices, real-time monitoring and control can be implemented remotely. The end sensors collect real-time data such as agricultural environment parameters, power consumption, biomedical data, and machine conditions and then send the data to remote servers. The servers receive and store the collected data, then extract and evaluate the data to provide the appropriate control measures. The end devices carry out control commands that are received from the server. There is a risk of the adversary controlling the communication channels and compromising the secret credentials of servers and end devices.

### 2.2. Elliptic Curve Encryption Mathematical Problems

Let $q > 3$ be a big prime number, $E(a, b)$ denote a non-singular elliptic curve over a finite field $F_q$, and $P$ be a generator point. The group operation is the usual multiplication of points on the elliptic curve, and $G$ is a subgroup of order $p$, where $p > q$ [39]. Hence, the following applies.

**Definition 1.** *Elliptic curve discrete logarithm (ECDL) problem: For the given points $X$ and $aX$, where $X \in G$ and $a \in Z_q^*$, it is computationally intractable to find $a$.*

**Definition 2.** *Elliptic curve Diffie–Hellman (ECDH) problem: For the given points $aX, bX \in G$, where $X \in G$ and $a, b \in Z_q^*$, finding point $abX$ is computationally intractable.*

### 2.3. Symbols

Symbols for the schemes are cataloged in Table 1.

**Table 1.** Symbols for the schemes.

| Notation | Description |
| --- | --- |
| $TA, KGC$ | Trusted Authority, Key Generation Center |
| $\mathcal{A}, \mathcal{C}$ | Adversary, Challenger |
| $SP_j, ID_{SP_j}$ | $j^{th}$ service provider and its identity |

**Table 1.** *Cont.*

| Notation | Description |
|---|---|
| $SM_i, ID_{SM_i}$ | $i^{th}$ smart meter and its identity |
| $E_q(a, b)$ | A non-singular elliptic curve |
| $P$ | A base point of $E_q(a, b)$ |
| $t, T_{pub}$ | Private-public key pair of *TA* [24] |
| $SK_{ij}, SSK_i$ | Session key |
| $\oplus, \|$ | Bitwise XOR and concatenation operations |
| $TS$ | Timestamps |
| $\Delta T$ | Maximum transmission delay |
| $h(\cdot)$ | One-way hash functions |
| $S, SP$ | End device, Server |
| $k/K$ | Private/public key of a entity |

## 3. Security Analysis of Srinivas et al.'s Scheme [24]

Srinivas et al. [24] put forward an AKA scheme for IoT smart grid systems with an Schnorr signature mechanism based on ECC. Before being added to the network, TA is responsible for distributing secret credentials, including signatures, to each smart grid and service provider. Smart meters and service providers can authenticate each other to establish session keys for secret communication. Baruah et al. [23] point out that the scheme of Srinivas et al. [24] is insured against MIM attacks and IM attacks. Cryptanalysis shows that the protocol [24] also suffers from key escrow issues and ESL attacks. For the review of Srinivas et al. [24], please refer to the complete paper.

### 3.1. Key Escrow Problem

During the registration process, TA generates the private keys of $SM_i$ and $SP_j$ with Schnorr's signature. *TA* calculates $T_{SM_i} = t_{SM_i} \cdot P$ and $M_{SM_i} = t_{SM_i} + h(T_{SM_i} \| ID_{SM_i}) \cdot t \pmod{q}$ for $SM_i$, and also $T_{SP_j} = t_{SP_j} \cdot P$, $P_{SP_j} = t_{SP_j} + h(T_{SP_j} \| ID_{SP_j}) \cdot t \pmod{q}$ for $SP_j$. Then, the long-term private secrets, $T_{SM_i}$, $M_{SM_i}$, $T_{SP_j}$, and $P_{SP_j}$, are known to him/her.

### 3.2. No Resistance to ESL Attacks

An AKA protocol is designed to resist an ESL attack, meaning that even if all the session-specific information of the entities in a session is compromised, the secrecy of the session key would remain uncompromised. During the authentication process, once the ephemeral secrets $r_i$ and $r_j$ are compromised, $\mathcal{A}$ can compromise the session key $SK_{ij}$ or $SK_{ji}$ by the following steps:

A1: $\mathcal{A}$ obtains the messages $MSG_1 = \{R_i, TS_i\}$, $MSG_2 = \{R_j, V_j, T_{SP_j}, TS_j\}$ and $MSG_3 = \{B_i, C_i, TS'_i\}$ by eavesdropping via the open channels;

A2: $\mathcal{A}$ extracts $TS_i, T_{SP_j}, TS_j, B_i$ and $TS'_i$ from the messages, then $\mathcal{A}$ calculates $S_i = h(r_i \| TS_i) \cdot (T_{SP_j} + h(T_{SP_j} \| ID_{SP_j}) \cdot T_{pub})$;

A3: For $S_i = S_j$, $\mathcal{A}$ gets $(ID_{SM_i} \| T_{SM_i}) = B_i \oplus h(S_i \| TS'_i)$ then calculates $U_j = h(r_j \| TS_j) \cdot (T_{SM_i} + h(T_{SM_i} \| ID_{SM_i}) \cdot T_{pub})$.

A4: For $A_i = U_j$, $\mathcal{A}$ calculates $SK_{ij} = h(A_i \| S_i \| ID_{SM_i} \| ID_{SP_j})$.

## 4. The Proposed Protocol

The proposal involves three phases: initialization, registration, and authentication and key agreement. To begin, TA generates and releases parameters for the system during the initialization phase. In the registration phase, each end device $S_s$ or server $SP_{sp}$ acquires its private key and both parties' public key with the assistance of *TA*. Ultimately, $S_s$ and $SP_{sp}$ will authenticate each other and negotiate a session key.

### 4.1. Initialization Phase

TA generates and releases parameters for the system as follows:

TA1: TA selects an elliptic curve $E(a, b)$ over finite field $F_q$ with a base point $P$;

TA2: Then, TA picks $h(\cdot)$ as the collision-resistant one-way hash function;

TA3: TA issues $\{(E(a, b), p, q, P, h(\cdot)\}$ publicly.

### 4.2. Registration Phase

As shown in Figure 2, taking the registration of the server $SP$ as an example, the processes are as follows:

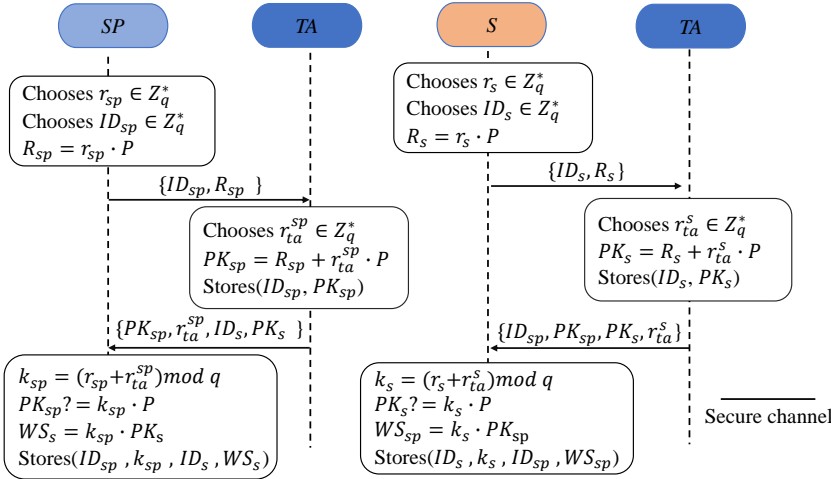

**Figure 2.** Registration processes of the proposed scheme.

R1: Firstly, SP chooses a random $r_{sp} \in Z_q^*$ and its identifier $ID_{sp} \in Z_q^*$ and computes $R_{sp} = r_{sp} \cdot P$. Then, SP transmits a registration request, $\{ID_{sp}, R_{sp}\}$, to TA securely.

R2: In response, first, TA chooses $r_{ta}^{sp} \in Z_q^*$ randomly to calculate the public key of $SP$. $PK_{sp} = R_{sp} + r_{ta}^{sp} \cdot P$. Next, TA sends $\{PK_{sp}, r_{ta}^{sp}, ID_s, PK_s\}$ to SP via a secure channel.

R3: In response, $SP$ takes $r_{ta}^{sp}$ as part of its private key and obtains its private key, $k_{sp} = ((r_{sp} + r_{ta}^{sp}) \bmod q)$. Then, SP checks whether $PK_{sp}? = k_{sp} \cdot P$; if it holds, then SP computes $WS_s = k_{sp} \cdot PK_s$ and stores $(ID_{sp}, k_{sp}, ID_s, WS_s)$.

Similarly, $S$ stores $(ID_s, k_s, ID_{sp}, WS_{sp})$ after registration. When a new end device $S'$ joins and registers the system, TA sends $\{ID_{s'}, PK_{s'}\}$ to SP securely.

### 4.3. Authentication and Key Agreement Phase

$S_s$ and $SP_{sp}$ will authenticate each other and negotiate a session key as shown in Figure 3.

S1: $S$ first picks $x_s \in Z_q^*$ randomly and generates a timestamp $TS_s$. Next, S calculates $A_s = (x_s k_s \bmod q) \cdot P$ and $B_s = x_s \cdot WS_{sp}$. Third, $S$ encrypts $ID_s$, $EID_s = ID_s \oplus B_s$, and obtains a verifier $V_s = h(WS_{sp} \| TS_s \| ID_s \| B_s)$. Finally, $S$ transmits the authentication request $M_s = \{A_s, EID_s, TS_s, V_s\}$ to $SP$.

SP1: Upon receiving the request message, $SP$ first examines its freshness against the timestamp $TS_s$. Next, SP calculates $B_{sp} = k_{sp} \cdot A_s$ to decrypt $ID_s = EID_s \oplus B_{sp}$. Thus, SP gains the S verifier and validates the equation of $V_s = ?h(WS_s \| TS_s \| ID_s \| B_{sp})$ to assure the integrity of the incoming message and the validity of S.

SP2: Firstly, $SP$ selects $x_{sp}$ randomly and obtains a timestamp $TS_{sp}$. Secondly, SP calculates $A_{sp} = (x_{sp} k_{sp} \bmod q) \cdot P$ and $C_{sp} = x_{sp} \cdot B_{sp}$. SP obtains the session key as $SSK_{sp} = h(ID_s \| ID_{sp} \| B_{sp} \| C_{sp})$. Third, SP figures out a verifier: $V_{sp} = h(WS_s \| TS_{sp} \| ID_{sp} \| SSK_{sp})$. and transmits authentication reply $M_{sp} = \{A_{sp}, TS_{sp}, V_{sp}\}$ to S.

S2: On receiving the message, S first examines its freshness against $TS_{sp}$. Next, S calculates $C_s = (x_s k_s \bmod q) \cdot A_{sp}$ to obtain the session key $SSK_s = h(ID_s \| ID_{sp} \| B_s \| C_s)$. Thus, S gains the SP verifier and validates the equation of $V_{sp} = ?h(WS_{sp} \| TS_{sp} \| ID_{sp} \| SSK_s)$ to assure the integrity of the incoming message and the validity of SP.

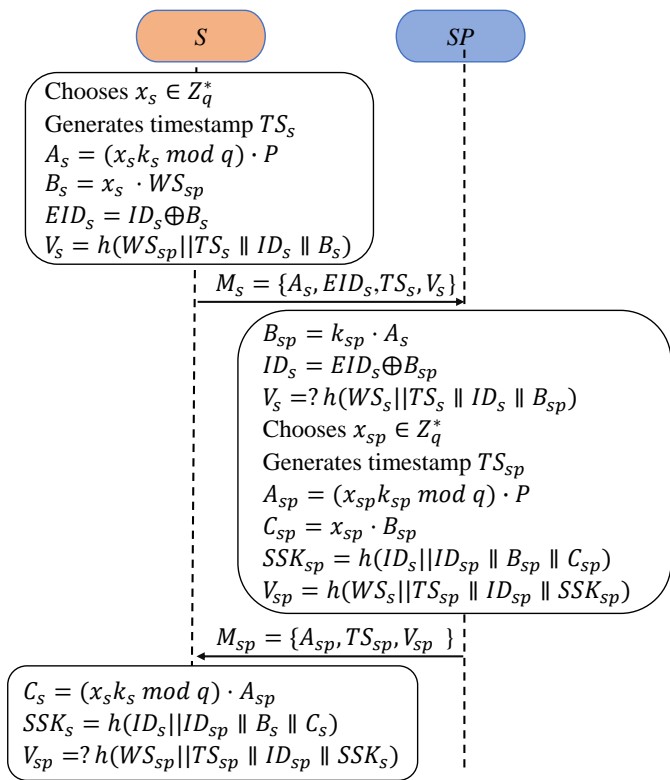

**Figure 3.** Authentication and key agreement of proposed protocol.

## 5. Security Analysis

This session provides a formal proof, descriptive security analysis and validation with ProVerif of the proposed scheme security.

### 5.1. Formal Proof

The eCK adversary model [33,34,40] is employed for the security proof. The system authentication model is shown in Figure 4. After registration, $SP_{sp}$ obtains its private and public keys $(k_{sp}, PK_{sp})$. The private and public keys of $S_s$ are $(k_s, PK_s)$. During authentication and key agreement, they negotiate the session key $SSK_s$ ($SSK_{sp}$) by exchanging authentication information $M_s$ and $M_{sp}$, where $M_s = f_s(x_s k_s)$ and $M_{sp} = f_s(x_{sp} k_{sp})$, $x_s$, and $x_{sp}$ are random ephemeral secrets.

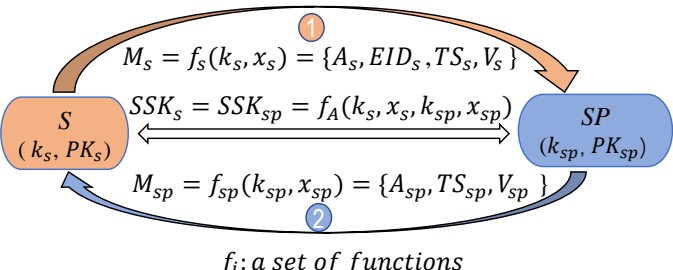

$$f_i: a \ set \ of \ functions$$

**Figure 4.** Authentication model.

#### 5.1.1. Security Model

**Participants**. There are $n$ participants in the proposed protocol $\mathbb{P}$, which are uniformly denoted by the set $F = \{F_1, \ldots, F_n\}$, and each participant may have $i$ instances involved in distinct, possibly concurrent executions of $\mathbb{P}$, where $n$ and $i$ are polynomial numbers.

**Sessions**. Let $\prod_{i,j}^{m}$ denote the $m$th protocol session running between entity $F_i$ and intended partner entity $F_j$. A session $\prod_{i,j}^{m}$ is *accepted* if it has computed a session key $SK_{i,j}^{m}$,

with a session identifier of $sid_{i,j}^m = (ID_i, ID_j, X_i, X_j)$, where $X_i$ is the outgoing information of $F_i$, and $X_j$ is the outgoing information of $F_j$.

**Adversary**. Firstly, the adversary $\mathcal{A}$ has complete control of the communicating network. Namely, $\mathcal{A}$ is able to eavesdrop on, alter, ascertain, and inject communication messages $M_s$ and $M_{sp}$. Secondly, $\mathcal{A}$ can have knowledge of the participant's long-term private key $k_s$ ($k_{sp}$) and ephemeral secret $x_s$ ($x_{sp}$) but not both. Thirdly, $\mathcal{A}$ allows replacing the participant's public key $PK_s$ ($PK_{sp}$). Finally, $\mathcal{A}$ can obtain the session key $SSK_s$ ($SSK_{sp}$) held by the participant. $\mathcal{A}$ can interact with $\prod_{i,j}^m$ with the following Oracle queries:

(1) $ESReveal(\prod_{i,j}^m)$. $\mathcal{A}$ can obtain the ephemeral secrets of $F_i$ with the query.

(2) $PKReplace(ID_i)$. $\mathcal{A}$ replaces the public key of $F_i$ using this query.

(3) $PKReveal(ID_i)$. $\mathcal{A}$ is available with this query for the public key of $F_i$.

(4) $SKReveal(ID_i)$. By running the query, $\mathcal{A}$ is able to obtain the long-term private keys of $F_i$, while the public key of $F_i$ has not yet been replaced.

(5) $SSKReveal(\prod_{i,j}^m)$. Returns $\perp$ if session $\prod_{i,j}^m$ was not *accepted*. If not, it returns the session key that $\prod_{i,j}^m$ holds.

(6) $Send(\prod_{i,j}^m, M)$. $\mathcal{A}$ represents $F_j$ sending the message $M$ to $F_i$ in session $\prod_{i,j}^m$ then receiving a reply from $F_i$ according to $\mathbb{P}$.

(7) $Test(\prod_{i,j}^m)$. The query does not simulate the adversary's ability, but it simulates the indistinguishability between real session keys and random keys. Input session $\prod_{i,j}^m$ must be fresh. As a challenger, $\mathcal{C}$ tosses a coin $b \in \{0, 1\}$. If $b = 0$, $\mathcal{C}$ returns the session key held by $\prod_{i,j}^m$; if $b = 1$, $\mathcal{C}$ returns a random key from the distribution of the session key.

**Matching session**. If $\prod_{i,j}^m$ and $\prod_{j,i}^n$ have the same session *sid*, then $\prod_{j,i}^n$ is said to be a matching session for $\prod_{i,j}^m$.

**Freshness**. Let $\prod_{i,j}^m$ denote an *accepted* session between honest participants $F_i$ and $F_j$ if $\prod_{i,j}^m$ and $\prod_{j,i}^n$ are matching sessions. $\prod_{i,j}^m$ is fresh if all the following conditions do not hold:

(1) $\mathcal{A}$ issues $SSKReveal(\prod_{i,j}^m)$ or $SSKReveal(\prod_{j,i}^n)$ queries if $\prod_{j,i}^n$ exists.

(2) The matching session $\prod_{j,i}^n$ exists. $\mathcal{A}$ makes $SKReveal(ID_i)$ and $ESReveal(\prod_{i,j}^m)$ queries, or $SKReveal(ID_j)$ and $ESReveal(\prod_{j,i}^n)$ queries.

(3) The matching session $\prod_{j,i}^n$ does not exist. $\mathcal{A}$ makes $SKReveal(ID_i)$ and $ESReveal(\prod_{i,j}^m)$, or $SKReveal(ID_j)$ queries.

A game simulates the security of an AKA protocol. In the game, $\mathcal{A}$ can issue multiple queries in any order. $\mathcal{A}$ can issue the $Test(\prod_{i,j}^m)$ query only once for a fresh session $\prod_{i,j}^s$. Next, a coin $b \in \{0, 1\}$ is flipped by $\mathcal{C}$. When the game ends, $\mathcal{A}$ will guess the value of $b$ as $b'$. If $b' = b$ and the test session $\prod_{i,j}^m$ is still fresh, then $\mathcal{A}$ wins the game. The advantage of $\mathcal{A}$ to win the game is defined as $Adv_{AKA}(A) = \left| \Pr[b' = b] - \frac{1}{2} \right|$.

**eCK Security**. To ensure the security of the AKA protocol in the eCK model, the following conditions must be met:

(1) If both parties complete a matching session, they will calculate the same session key, unless the probability is negligible.

(2) For any polynomial-time adversary $\mathcal{A}$, the advantage in breaking the AKA protocol, $Adv_{AKA}(\mathcal{A})$, must be negligible.

### 5.1.2. Formal Security Analysis

At first, three empty lists are created to hold the query and the corresponding answers.

$L$: input–output pairs of the hash function. Instead of being randomly chosen by $\mathcal{C}$, the real hash function computes the outputs. To complete the safety proof, $\mathcal{C}$ needs to record the mapping between the inputs and outputs.

$L_U$: Tuple $(ID_i, k_i, PK_i)$ for storing the queries–answers of $PKReveal(ID_i)$, $PKReplace(ID_i)$, and $SKReveal(ID_i)$.

$L_w$: Tuple $(ID_i, ID_j, s, x_i, x_j)$ for storing the queries–answers of $ESReveal(\prod_{i,j}^s)$.

To continue, it is essential to clarify a few fundamental configurations. Suppose that $\mathcal{A}$ is activating no more than $n_1$ honest parties, and each party is engaged in no more than $n_2$

sessions. Assume that $\mathcal{A}$ selects the $\prod_{I,J}^S$ as the test session. $\mathcal{A}$ can distinguish a test session key from a random string in the three ways below:

**A1. Guessing.** $\mathcal{A}$ guesses the session key correctly.

**A2. Key replication.** $\mathcal{A}$ creates a mismatched session that has the same session key as $\prod_{I,J}^S$. So $\mathcal{A}$ is able to fetch the session key by querying the mismatched session.

**A3. Forging.** The value of $h(ID_i\|ID_j\|B_i\|C_i)$ is computed at some point by $\mathcal{A}$.

**Theorem 1.** *Since the* ECDL *or* ECDH *problem is intractable, the advantage of $\mathcal{A}$ against the AKA scheme in the eCK model is negligible.*

**Proof.** Since the session key $SSK_i \in Z_q^*$, there is only a $\frac{1}{q-1}$ chance of guessing the correct $SSK_i$ in the **guessing** attack.

The hash function should yield the same results for different input values in order to prevent the **key replication** attack. The probability of success of a **key duplication** attack is negligible.

The analysis of the **forging attack** is shown below.

Consider the tuple $(P, u_1P, u_1u_2P, v_1P, v_1v_2P)$ as an example of the ECDH problem, in which the ephemeral keys $x_s$ and $x_{sp}$ are denoted by $u_2$ and $v_2$, and the long-term keys $k_s$ and $k_{sp}$ are represented by $u_1$ and $v_1$. If $\mathcal{A}$ is successful in **forging attack** with non-negligible probability, $\text{ECDH}(u_1u_2P, v_1P) = u_1u_2v_1v_1P$ and $\text{ECDH}(u_1u_2P, v_1v_2P) = u_1u_2v_1v_1P$ can be computed by $\mathcal{C}$ using $\mathcal{A}$.

First, $\mathcal{C}$ creates a test session $\prod_{I,J}^S$ by randomly selecting $S \in \{1, n_2\}$ and $I, J \in \{1, n_1\}(I \neq J)$. Therefore, $\mathcal{C}$ has no higher chance of correctly guessing the test session $\prod_{I,J}^S$ than $\frac{1}{n_1^2 \cdot n_2}$. Let $\prod_{J,I}^E$ be the matching session of $\prod_{I,J}^S$. There are six complementary events to consider as shown in Table 2. E1: $\mathcal{A}$ does not obtain the ephemeral secret keys of $ID_I(u_2)$ and $ID_J(v_2)$. E2: $\mathcal{A}$ does not obtain the ephemeral secret key of $ID_I(u_2)$ and secret value of $ID_J(v_1)$. E3: $\mathcal{A}$ does not obtain the ephemeral secret keys $ID_J(v_2)$ and secret value of $ID_I(u_1)$. E4: $\mathcal{A}$ does not obtain the secret values of $ID_I(u_1)$ and $ID_I(v_1)$. E5: There is no matching session for $\prod_{I,J}^S$. $\mathcal{A}$ obtains parameters similar to E2. E6: There is no matching session for $\prod_{I,J}^S$. $\mathcal{A}$ obtains parameters similar to E4.

**Table 2.** Complementary events.

|  | E1 | E2 | E3 | E4 | E5 | E6 |
|---|---|---|---|---|---|---|
| $\prod_{J,I}^E$ |  |  |  |  | × | × |
| Ephemeral secret keys of $ID_I(u_2)$ | × | × |  |  | × |  |
| Ephemeral secret keys of $ID_J(v_2)$ | × |  | × |  |  |  |
| Secret value of $ID_I(u_1)$ |  |  | × | × |  | × |
| Secret value of $ID_I(v_1)$ |  | × |  | × | × | × |

×: the session does not exit or $\mathcal{A}$ does not obtain the parameter.

At least one event in the set, $\{E1 \wedge A3, E2 \wedge A3, E3 \wedge A3, E4 \wedge A3, E5 \wedge A3, E6 \wedge A3\}$, happens with non-negligible probability if $\mathcal{A}$ succeeds in faking attack with non-negligible probability.

i.   **Analysis of E1**

(1)   **Setup**. $\mathcal{C}$ sends $(E(a, b), p, q, P, P, h(\cdot))$ to the $\mathcal{A}$.

(2)   **Query**. $\mathcal{A}$ will query the public key before an identity is used in any other queries, and all queries are different. $\mathcal{C}$ answers the queries issued by $\mathcal{A}$ as follows:

(1)   *PKReveal*$(ID_i)$. $\mathcal{A}$ submits an identity $ID_i$, $\mathcal{C}$ picks at random $k_i \in Z_q^*$, computes $PK_i = k_i \cdot P$, then returns $PK_i$ and adds $(ID_i, k_i, PK_i)$ to the list $L_U$.

**(2)** *PKReplace*($ID_i$). $\mathcal{A}$ submits a tuple $PK'_i = k'_i \cdot P$ for $ID_i$, $\mathcal{C}$ replaces $PK_i$ with $PK'_i$, and update $(ID_i, k_i, PK_i)$ with $(ID_i, *, K'_i)$ in the list $L_U$, where $*$ can be the secret value $k'_i$ or be the symbol $\perp$.

**(3)** *SKReveal*($ID_i$). $\mathcal{A}$ submits an identity $ID_i$, $\mathcal{C}$ looks up $(ID_i, k_i, PK_i)$ in the list $L_U$ and returns $k_i$. If $\mathcal{A}$ has replaced the public key $PK_i$ and has not submitted a new one, $\mathcal{C}$ will refuse to respond.

**(4)** *ESReveal*($\prod_{i,j}^{m}$). $\mathcal{A}$ submits a session $\prod_{i,j}^{s}$, then $\mathcal{C}$ processes as follows:

- If $\prod_{i,j}^{s} = \prod_{I,J}^{S}$ or $\prod_{i,j}^{s} = \prod_{J,I}^{E}$, then $\mathcal{C}$ fails and stops.
- If not, $\mathcal{C}$ selects $x_i, x_j \in Z_q^*$ at random and appends $(ID_i, ID_j, s, x_i, x_j)$ to $L_W$.

**(5)** *SSKReveal*($\prod_{i,j}^{m}$). $\mathcal{A}$ submits a session $\prod_{i,j}^{s}$, and $\mathcal{C}$ processes as follows: If $\mathcal{A}$ has replaced the public key $PK_i$ (or $PK_j$) and did not submit the new secret value $PK'_i$ (or $PK'_j$), then $\mathcal{C}$ may refuse to reply, else

*Case* 1 : If $\prod_{i,j}^{s} = \prod_{I,J}^{S}$ or $\prod_{i,j}^{s} = \prod_{J,I}^{E}$, then $\mathcal{C}$ fails and stops.
*Case* 2 : If $\mathcal{A}$ has made *ESReveal*($\prod_{i,j}^{m}$) for $\prod_{i,j}^{s}$, C will look up $(ID_i, ID_j, s, x_i, x_j)$ in $L_W$, $(ID_i, k_i, PK_i)$, or $(ID_j, k_j, PK_j)$ in $L_U$, then figures out the session key according to the AKA scheme.
*Case* 3 : Else, $\mathcal{C}$ selects $x_i, x_j \in Z_q^*$ at random and appends $(ID_i, ID_j, s, x_i, x_j)$ to $L_W$, then proceeds as in case 2.

**(6)** *Send*($\prod_{i,j}^{s}, M$). $\mathcal{C}$ will answer the query as below.

- If $(\prod_{i,j}^{s}, M) = (\prod_{I,J}^{S}, \perp)$, $\mathcal{C}$ looks up $(ID_I, k_I, PK_I)$ in $L_U$ and then returns $k_I u_2 P$.
- If $(\prod_{i,j}^{s}, M) = (\prod_{J,I}^{E}, \perp)$, $\mathcal{C}$ looks up $(ID_J, k_J, PK_J)$ in $L_U$ and then returns $k_I v_2 P$.
- If $\prod_{i,j}^{s} \neq \prod_{I,J}^{S}$ and $\prod_{i,j}^{s} \neq \prod_{J,I}^{E}$, $\mathcal{C}$ looks up $(ID_i, k_i, PK_i)$ in $L_U$ and processes as follows:
  - If $\mathcal{A}$ has made *ESReveal*($\prod_{i,j}^{m}$) for $\prod_{i,j}^{s}$, $\mathcal{C}$ looks up $(ID_i, ID_j, s, x_i, x_j)$ in $L_W$, then computes and returns $A_i$.
  - If not, $\mathcal{C}$ randomly selects $x_i, x_j \in Z_q^*$ and calculates and returns $A_i$, then appends $(ID_i, ID_j, s, x_i, x_j)$ to $L_W$.
- If $M = (A_j, *)$, $\mathcal{C}$ accepts $\prod_{i,j}^{s} \neq \prod_{I,J}^{S}$.

**(7)** *Test*($\prod_{i,j}^{s}$). If the public key $PK_i$ (or $PK_j$) had been replaced with $k'_i$ (or $k'_j$), $\mathcal{A}$ would have had to commit the new secret value $k'_i$ (or $k'_j$) to $\mathcal{C}$; since $\mathcal{C}$ is unable to generate the session key if he does not know the secret values for $ID_i$ and $ID_j$. The responses of $\mathcal{C}$ to *Test*($\prod_{i,j}^{s}$) are as follows:

- If $\prod_{i,j}^{s} \neq \prod_{I,J}^{S}$, C fails and stops.
- If $\prod_{i,j}^{s} = \prod_{I,J}^{S}$, $\mathcal{C}$ randomly chooses $SSK_i \in Z_q^*$ and sends it back to $\mathcal{A}$.

**(3)** **Solve ECDH problems**. To win the game by forging attack, $\mathcal{A}$ would have to calculate $h(ID_I \| ID_J \| B_I \| C_I)$, where $B_I = k_J k_I u_2 P$ and $D_I = k_J k_I u_2 v_2 P$. $\mathcal{C}$ finds $k_I$ and $k_J$ in $L_U$ and computes $B_I$ and $D_I$ by solving the ECDH problem.

**(4)** **Probability**. If it is possible for $\mathcal{C}$ to properly guess the test session $\prod_{I,J}^{S}$, $\mathcal{C}$ will not fail in the query phase. Thus, $\mathcal{C}$ is able to calculate $B_I = \text{ECDH}(k_J P, k_I u_2 P)$ and $D_I = \text{ECDH}(k_J v_2 P, k_I u_2 P)$ with probability $\frac{1}{n_1^2 n_2} Adv_{AKA}(A)$ if $\mathcal{A}$ wins in the game with advantage $Adv_{AKA}(A)$.

**ii.** **Analysis of E2**

**(1)** **Setup**. Same as that in the analysis of E1.

**(2)** **Query**. $\mathcal{C}$ responds to the queries from $\mathcal{A}$ as those in the analysis of E1 except for the $PKReveal(ID_i)$, $SKReveal(ID_i)$, $ESReveal(\prod_{i,j}^m)$, and $Send(\prod_{i,j}^s, M)$.

    **(1)** $PKReveal(ID_i)$. $\mathcal{A}$ submits an identity $ID_k$, $\mathcal{C}$ will respond to the query as follows:

- If $ID_k = ID_J$, $\mathcal{A}$ computes $K_J = v_1 P$, returns $v_1 P$, and adds $(ID_J, \bot, v_1 P)$ to the list $L_U$.
- If not, $C$ randomly selects $k_k \in Z_q^*$ and calculates $PK_k = k_k P$, then returns $PK_k$ and adds $(ID_k, k_k, PK_k)$ in $L_U$.

    **(2)** $SKReveal(ID_i)$. If $ID_i = ID_J$, $\mathcal{C}$ will fail and stop. If not, $\mathcal{C}$ looks up $(ID_i, k_i, PK_i)$ in $L_U$ and returns $k_i$.

    **(3)** $ESReveal(\prod_{i,j}^m)$. $\mathcal{C}$ will respond to the query as follows:

- If $\prod_{i,j}^s = \prod_{I,J}^S$ or $\prod_{i,j}^s = \prod_{J,I}^E$, $\mathcal{C}$ randomly chooses $x_J \in Z_q^*$ and returns $(\bot, x_J)$, then appends $(ID_J, ID_J, s, \bot, x_J)$ to $L_W$.
- If not, $\mathcal{C}$ randomly chooses $x_i, x_j \in Z_q^*$ and returns $(x_i, x_j)$, then appends $(ID_i, ID_j, s, x_i, x_j)$ to $L_W$.

    **(4)** $Send(\prod_{i,j}^s, M)$. $\mathcal{C}$ will respond to the query as follows:

- If $(\prod_{i,j}^s, M) = (\prod_{I,J}^S, \bot)$, $\mathcal{C}$ looks up $(ID_I, k_I, PK_I)$ in $L_U$ and returns $(k_1 u_2 P)$.
- If $(\prod_{i,j}^s, M) = (\prod_{J,I}^E, \bot)$, $\mathcal{C}$ looks up $(ID_J, \bot, v_1 P)$ in $L_U$, and $(ID_I, ID_J, S, \bot, x_J)$ in $L_W$, then sends $(v_1 x_J P)$ back.
- Otherwise, the analysis is the same as for E1.

**(3)** **Solve ECDH problems**. To win the game by forging attack, $\mathcal{C}$ must compute $h(ID_I \| ID_J \| B_I \| C_I)$, where $B_I = k_J k_I u_2 P$ and $C_I = k_I u_2 v_1 x_J P$. $\mathcal{C}$ finds $k_I$ in the list $L_U$ and $(\bot, x_J)$ in the list $L_W$ to compute $B_I$ and $C_I$ by solving ECDH problems.

**(4)** **Probability**. If it is possible for $\mathcal{C}$ to properly guess the test session $\prod_{I,J}^S$, $\mathcal{C}$ will not fail in the query phase. Thus, $\mathcal{C}$ is able to calculate $B_I = \text{ECDH}(k_J P, k_I u_2 P)$ and $D_I = \text{ECDH}(v_1 x_J P, k_I u_2 P)$ with the same probability as E1 winning the game.

**iii. Analysis of E3**

$\mathcal{C}$ can swap $ID_I$ and $ID_J$ in E3 and then carry out the analysis of E2.

**iv. Analysis of E4**

**(1)** **Setup**. This is the same as that in the analysis of E1.

**(2)** **Query**. The responses of $\mathcal{C}$ to the queries from $\mathcal{A}$ are the same as in E1, except for $PKReveal(ID_i)$, $SKReveal(ID_i)$, $ESReveal(\prod_{i,j}^m)$, $SKReveal(ID_i)$, and $Send(\prod_{i,j}^s, M)$ queries.

    **(1)** $PKReveal(ID_i)$. $\mathcal{A}$ submits an identity $ID_k$, $\mathcal{C}$ process as follows:

- If $ID_k = ID_I$, $\mathcal{C}$ computes $K_I = u_1 P$, then returns $u_1 P$ and appends $(ID_I, \bot, u_1 P)$ to $L_U$.
- If $ID_k = ID_J$, $\mathcal{C}$ computes $K_J = v_1 P$, then returns $v_1 P$ and appends $(ID_J, \bot, v_1 P)$ to $L_U$.
- Else, $C$ chooses $k_k \in Z_q^*$ randomly and calculates $K_k = k_k P$, then returns $K_k$ and adds $(ID_k, k_k, K_k)$ in $L_U$.

    **(2)** $SKReveal(ID_i)$. If $ID_i = ID_I$ or $ID_i = ID_J$, then $\mathcal{C}$ fails and stops. If not, $C$ looks up $(ID_i, k_i, K_i)$ in $L_U$ and returns $k_i$.

    **(3)** $ESReveal(\prod_{i,j}^m)$. $\mathcal{A}$ submits a session $\prod_{i,j}^s$, $\mathcal{C}$ randomly chooses $x_i, x_j \in Z_q^*$ and returns $(x_i, x_j)$, then appends $(ID_i, ID_j, s, x_i, x_j)$ to $L_W$.

    **(4)** $Send(\prod_{i,j}^s, M)$. $\mathcal{C}$ finds $(ID_i, k_i, K_i)$ in the list $L_U$, then responds to queries as follows:

- If $(\prod_{i,j}^s, M) = (\prod_{I,J}^S, \perp)$, $\mathcal{C}$ performs as follows:
  - If $\mathcal{A}$ has made $ESReveal(\prod_{i,j}^m)$ for $\prod_{i,j}^s$, $\mathcal{C}$ looks up $(ID_i, ID_j, s, x_i, x_j)$ in $L_W$ and returns $(u_1 x_i P)$.
  - If $\mathcal{A}$ has made $ESReveal(\prod_{j,i}^m)$ for $\prod_{j,i}^s$, $\mathcal{C}$ looks up $(ID_i, ID_j, s, x_i, x_j)$ in $L_W$ and returns $(v_1 x_j P)$.
  - Else, $\mathcal{C}$ randomly chooses $x_i, x_j \in Z_q^*$ and returns $A_i$, then appends $(ID_i, ID_j, s, x_i, x_j)$ to $L_W$.
- $M = (A_j, *)$, $\mathcal{C}$ accepts the session.

**(3)** **Solve ECDH problems.** To win the game by forging attack, $\mathcal{C}$ must compute $h(ID_I \| ID_J \| B_I \| D_I)$, where $B_I = u_1 v_1 x_I P$ and $D_I = u_1 x_I v_1 x_J P$. $\mathcal{C}$ looks up $(ID_i, ID_j, s, x_i, x_j)$ in $L_W$ to compute $B_I$ and $D_I$ by solving ECDH$_1$ and ECDH$_2$ problems.

**(4)** **Probability.** If it is possible for $\mathcal{C}$ to properly guess the test session $\prod_{I,J}^S$, $\mathcal{C}$ will not fail in the query phase. Thus, $\mathcal{C}$ is able to calculate $B_I = \text{ECDH}_1(v_1 P, u_1 x_I P)$ and $D_I = \text{ECDH}_2(u_1 x_I P, v_1 x_J P)$ with the same probability as E1 winning the game.

**v.** **Analysis of E5**

In E2, there is a matching session $\prod_{J,I}^E$ for the test session $\prod_{I,J}^S$, whereas in E5, there is no matching session for $\prod_{I,J}^S$. Therefore, the analysis for E5 is similar to that for E2.

**vi.** **Analysis of E6**

In E4, there is a matching session $\prod_{J,I}^E$ for the test session $\prod_{I,J}^S$. However, in E6, there is no matching session for $\prod_{I,J}^S$. Therefore, the analysis of E6 is similar to that of E4.

$\square$

*5.2. Descriptive Security Analysis*

5.2.1. No Key Escrow Issues

During registration, S obtains the long-term private key, $k_s = (r_s + r_{ta}^s) \bmod q$. TA only generates the partial private key $r_{ta}^s$, which avoids the key escrow problems. The long-term private key of SP is similar.

5.2.2. ESL Attack Resistance

Resistance to ESL attacks means $\mathcal{A}$ is unable to figure out the session key in spite of knowing ephemeral secrets $x_s$ and $x_{sp}$. For $SSK_s = H(ID_s \| ID_{sp} \| B_s \| C_s)$, where $C_s = (x_s k_s \bmod q) \cdot A_{sp} = (x_{sp} k_{sp} \bmod q) \cdot A_s$, even if $x_s$ and $x_{sp}$ are revealed, $\mathcal{A}$ cannot figure out $SSK_s$ because they do not know the long-term secrets $k_s$ and $k_{sp}$. Similarly, if $\mathcal{A}$ knows the short-term secrets $x_s$ and $x_{sp}$, then he/she cannot calculate $SSK_{sp}$.

5.2.3. Anonymity

In this scheme, $ID_s$ and $ID_{sp}$ are masked before being transmitted during the authentication process and change dynamically from session to session with the choice of the temporary random numbers $x_s$ and $x_{sp}$. $\mathcal{A}$ is incapable of retrieving and tracing the identity from the transmitted messages. That is, the proposal guarantees anonymity.

5.2.4. Mutual Authentication

During authenticating, $S$ verifies $SP$ by checking the correctness of $V_{sp}$. For $V_{sp} = h(WS_s \| TS_{sp} \| ID_{sp} \| SSK_{sp})$, where $SSK_{sp} = x_{sp} \cdot B_{sp} = k_{sp} x_{sp} \cdot A_s$, $V_{sp}$ cannot be figured out without long-term secrets $k_{sp}$ of $SP$. Similarly, $SP$ verifies $S$ by checking $V_s$.

5.2.5. Impersonation Attacks Resistance

Firstly, we analyze the S impersonation attack. If $\mathcal{A}$ tries to impersonate $S$ to generate the message $\{A_s, EID_s, TS_s, V_s\}$ to make $SP$ believe that the message is legitimate and

generated by $S$, $\mathcal{A}$ cannot generate valid information and impersonate S in polynomial time without knowing parameters such as $k_s$ and $x_s$.

### 5.2.6. IoT Nodes Capture Attack Resistance

Some IoT end devices are placed in unattended environments and may be physically captured by an adversary. Thus, their credentials $\{ID_s, k_s, PK_s, ID_{sp}, Ws_{sp}\}$ can be easily extracted by $\mathcal{A}$. The credentials for different end devices in the proposed scheme are different. Therefore, this will only lead to session key leakage between the captured $S_s$ and the server $SP$ but not between the uncorrupted end device $S'_s$ and the server $SP$. This implies that the proposal can withstand IoT node capture attacks.

### 5.2.7. KCI Attack Resistance

Resistance against KCI attacks refers to the inability of $\mathcal{A}$ to impersonate another legitimate participant, Bob, to authenticate with Alice after Alice's long-term private key disclosure. Suppose $\mathcal{A}$ learns the long-term key $k_s$ of the end device $S$ and wants to impersonate $SP$ to produce $\{A_{sp}, TS_{sp}, V_{sp}\}$ to convince $S$ that the message is legitimate and generated by $SP$. For $V_{sp} = h(WS_s \| TS_{sp} \| ID_{sp} \| SSK_{sp})$, where $C_{sp} = x_{sp} \cdot B_{sp} = k_{sp}x_{sp} \cdot A_s$, and $k_{sp}$ has not been compromised, $\mathcal{A}$ cannot impersonate server $SP$ to perform authentication and key agreement with $S$. Similarly, $\mathcal{A}$ cannot carry out KCI attacks against $SP$.

### 5.3. Automatic Formal Verification

The security of the proposal is formally validated with ProVerif [5]. Table 3 illustrates the codes of $S$, where *schs* is a secret channel used for $S$ registration, and *ch* is a public channel used for $S$ and $SP$ authentication. Based on the following results, it can be concluded that both the authentication process and the session key are secure from adversary attacks.

**Table 3.** Codes for end device $S$.

```
let S =
new rs:bitstring;
let Rs = Mul(rs, P) in
out (schs, (IDs, Rs));
in (schs, (vIDsp:bitstring,vPKsp: bitstring, vPKs: bitstring,vrtas:bitstring));
let ks = add (rs, vrtas) in
let PKs= Mul (ks, P) in
if PKs = vPKs then
let WSsp = Mul (ks, vPKsp) in
!
(
event startAuthsp;
let As = Mul(xs,PKs) in
let Bs = Mul(xs, WSsp) in
let EIDs = xor (IDs, Bs) in
new TSeeds:bitstring;
let Ts = generate_Timeline(TSeeds) in
let Vs = Hash(con (con (con (WSsp, Ts), IDs),Bs))in
out (ch, (As, EIDs, Ts, Vs));
in (ch, (vAsp: bitstring, vTsp: bitstring, vVsp:bitstring));
let Cs = Mul (mul (xs, ks), vAsp) in
let SSKs = Hash(con (con (con (IDs,IDsp), Bs), Cs)) in
let Vsp = Hash(con (con (con (WSsp,vTsp),IDsp), SSKs)) in
if Vsp = vVsp then
event endAuths;
0
).
```

Here are the results of the queries in ProVerif:

(1)   RESULT inj-event(endAuthS) ==> inj-event(startAuthS) is true.
(2)   RESULT inj-event(endAuthSP) ==> inj-event(startAuthSP) is true.
(3)   RESULT inj-event(endAuthSP) ==> inj-event(endAuthS) is true.
(4)   RESULT inj-event(endAuthS) ==> inj-event(endAuthSP) is true.
(5)   RESULT not attacker(SSKs[]) is true.
(6)   RESULT not attacker(SSKsp[]) is true.
(7)   RESULT not attacker(ks[]) is true.
(8)   RESULT not attacker(ksp[]) is true.

## 6. Performance Comparison

### 6.1. Communication Cost

According to [22,41], suppose that $G_1$ is an additive cyclic group with order $q_1$. $G_2$ is a multiplicative cyclic group with order $q$. The bilinear map is defined as $e : G_1 \times G_1 \rightarrow G_2$. In addition, it is assumed that the lengths of an identifier (ID), a hash output (H), a timestamp (TS), and a random number (R) are 64, 128, 32, and 128 bits, respectively. Table 4 shows the communication overhead of each protocol during the authentication and key negotiation phases. It can be concluded that the proposed scheme has the lowest communication overhead in the authentication and key negotiation processes.

**Table 4.** Communication cost.

| Scheme | End Device (bit) | Server (bit) | Total (bit) |
|---|---|---|---|
| [22] | $2G + G1 + H + TS + ID = 2016$ | $G + H + TS = 544$ | 2560 |
| [24] | $2G + H + 2TS + ID = 1024$ | $2G + H + TS = 928$ | 1952 |
| [25] | $2G + 2H + TS = 1056$ | $G + H + TS = 512$ | 1568 |
| [27] | $G + H + R + 2TS + ID = 832$ | $G + H + 2TS + ID = 640$ | 1472 |
| [28] | $G + 2H + TS = 672$ | $G + 2H + ID = 704$ | 1376 |
| [29] | $3G + 2H + ID = 1472$ | $3G + H + ID = 1344$ | 2816 |
| Ours | $G + H + TS + ID = 608$ | $G + H + TS = 544$ | 1152 |

### 6.2. Computation Cost

According to He et al. [41], Table 5 shows the run-time of the relevant encryption operation on a *Samsung Galaxy S*5. Table 6 displays the run-time of each scheme during authentication and key agreement. It is evident that the proposed scheme requires the least computational overhead.

**Table 5.** Run-time of related operations.

| Notation | Operation | Time (ms) |
|---|---|---|
| $T_{bp}$ | Bilinear pairing | 32.713 |
| $T_h$ | Hash function | 0.006 |
| $T_{pm1}$ | Point multiplication in G1 | 13.405 |
| $T_{pa1}$ | Point addition in G1 | 0.56 |
| $T_{exp2}$ | Exponentiation in G2 | 2.249 |
| $T_s$ | Symmetric encryption | 0.012 |
| $T_{pa}$ | ECC point addition | 0.014 |
| $T_{pm}$ | ECC point multiplication | 3.352 |

**Table 6.** Computation cost.

| Scheme | End Device (ms) | Server (ms) | Total (ms) |
|---|---|---|---|
| [22] | $2T_{pm1} + T_{pa1} + T_{exp2} + 4T_{pm} + T_{pa} + 6T_h = 43.077$ | $T_{pb} + 4T_{pm} + T_{pa} + 5T_h = 46.165$ | 89.242 |
| [24] | $4T_{pm} + T_{pa} + 7T_h = 13.464$ | $4T_{pm} + T_{pa} + 7T_h = 13.464$ | 26.982 |
| [25] | $3T_{pm} + 4T_h = 10.08$ | $3T_{pm} + 2T_{pa} + 5T_h = 13.466$ | 23.546 |
| [27] | $3T_{pm} + 2T_s + 4T_h = 10.104$ | $4T_{pm} + 3T_s + 4T_h = 13.468$ | 23.572 |
| [28] | $4T_{pm} + 7T_h = 13.45$ | $4T_{pm}7T_h = 13.45$ | 26.9 |
| [29] | $7T_{pm} + 2T_{pa} + 5T_h = 23.522$ | $7T_{pm} + 2T_{pa} + 5T_h = 23.522$ | 47.044 |
| Ours | $3T_{pm} + 3T_h = 10.074$ | $3T_{pm} + 3T_h = 10.074$ | 20.148 |

*6.3. Performance Comparison*

The results of the comparison between the proposal and related schemes [22,24,25,27–29] in terms of security are shown in Table 7. Compared to the existing schemes, the proposed protocol provides better security and functionality, e.g., it is resistant to attacks such as IM, MIM, and ESL while providing anonymity, mutual authentication, and PFS without key escrow issues.

**Table 7.** Performance comparison.

| Scheme | SF1 | SF2 | SF3 | SF4 | SF5 | SF6 | SF7 | SF8 | SF9 | SF10 | SF11 | SF12 |
|--------|-----|-----|-----|-----|-----|-----|-----|-----|-----|------|------|------|
| [22] | × | × | √ | × | √ | √ | × | √ | √ | × | √ | √ |
| [24] | × | × | √ | × | √ | √ | √ | √ | √ | √ | √ | × |
| [25] | √ | √ | √ | √ | √ | √ | × | √ | √ | √ | √ | × |
| [27] | √ | × | √ | √ | √ | √ | √ | √ | √ | √ | √ | × |
| [28] | √ | × | √ | √ | × | √ | √ | √ | √ | × | √ | √ |
| [29] | √ | √ | √ | × | √ | √ | √ | √ | × | √ | √ | √ |
| Ours | √ | √ | √ | √ | √ | √ | √ | √ | √ | √ | √ | √ |

SF1: IM attack resistance; SF2: MIM attack resistance; SF3: Mutual authentication without the help of TA; SF4: ESL attack resistance; SF5: KCI attack; SF6: IoT nodes capture attack resistance; SF7: Anonymity; SF8: Unknown key share attack resistance; SF9: Perfect forward secrecy; SF10: Formal security proof; SF11: Replay attack resistance; SF12: No key escrow issue; √: Secure or supportive ×: Insecure or unsupported.

## 7. Conclusions

To begin, we reviewed the existing ECC-based AKA schemes. Then, we pointed out that the existing schemes failed to provide user anonymity and PFS and had no resistance to typical attacks (such as ESL, IM, MIM, KCI, etc.) with key escrow problems. The high computational and communication costs also made some of these solutions unsuitable for resource-limited IoT. Furthermore, the security proofs were conducted in the Random-Oracle model. It is widely recognized that cryptographic schemes proven secure in the Random-Oracle model may not necessarily provide the same level of security when implemented in real-world systems. We propose a security-enhanced AKA protocol for connecting IoT devices to servers to remedy the existing challenges. The session key security of the proposed scheme is rigorously proven under the eCK model with the elliptic curve encryption computational assumptions. The session key confidentiality and authentication properties are verified with ProVerif. Based on the performance comparison, it is found that the proposed scheme offers user anonymity, PFS, mutual authentication, and resistance to typical attacks such as ESL, IM, MIM, and KCI. Additionally, the proposed scheme has minimal computational and communication overhead compared to the existing schemes.

**Author Contributions:** Conceptualization, S.H., S.J. and Q.M.; methodology, S.H. and F.Y.; software, W.Z.; validation, S.H. and P.D.; formal analysis, S.H. and S.J.; investigation, S.H., S.J. and Q.M.; resources, S.H.; data curation, Q.M.; writing—original draft preparation, S.H. and F.Y.; writing—review and editing, S.H., S.J., Q.M., F.Y., W.Z. and P.D.; visualization, S.H.; supervision, S.H.; project administration, S.H.; funding acquisition, S.H. All authors have read and agreed to the published version of the manuscript.

**Funding:** This research was supported in part by the Natural Science Foundation of China (No. 62072319).

**Data Availability Statement:** The data presented in this study are available on request from the corresponding author. The data are not publicly available due to privacy.

**Conflicts of Interest:** The authors declare no conflicts of interest.

## Abbreviations

The following abbreviations are used in this manuscript:

| | |
|--------|--------|
| IoT | Internet of Things |
| AKA | Authentication key agreement |
| TA | Trusted authority |

| | |
|---|---|
| ECC | Elliptic Curve Cryptography |
| PUF | Physical Unclonable Function |
| CRP | Challenge–Response Pair |
| BR | Bellare and Rogaway |
| mBR | Modified BR model |
| CK | Canetti and Krawczyk |
| WBAN | Wireless body area networks |
| WSN | Wireless sensor networks |
| SG | Smart grid |
| PFS | Perfect forward security |
| IM | Impersonation |
| KCI | Key compromise impersonation |
| MIM | Man-in-the-middle |
| ESL | Ephemeral secret leakage |

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
