# Peer review of "Provably Secure ECC-Based Anonymous Authentication and Key Agreement for IoT"

_applsci, doi:10.3390/app14083187_

Round 1
Reviewer 1 Report
Comments and Suggestions for Authors
Dear Authors,
Please find the following comments on your paper.
1. Please add result details in the abstract.
2. Please add a full stop at the end of the figure captions.
3. Tab 1 Should change to Table 1.
4. Please add a detailed explanation to the Table contents.
Regards
Comments on the Quality of English LanguageMinor editing of English language required
Author Response
Dear Reviewer,
We are very grateful to you for your constructive comments on our submission, applsci-2852529. We have taken all of them into account when preparing the current revision and this response letter. Please find below detailed responses to your comments. We cite your different types of comments, followed by indices, and give our answers and actions to the respective comments. We mark the answered content with a red color and the modified content with a blue color in the revised manuscript. Moreover, we also put the revision in this letter for the respective comments.
We hope the following detailed responses can help the editor and the reviewers better understand our revised manuscript. Thanks for your consideration! We are looking forward to your further comments.
Sincerely yours,
Shunfang Hu
On behalf of the authors

Reviewer 2 Report
Comments and Suggestions for Authors
The authors focused on the Internet of Things security issues. They presented a new authentication key agreement protocol for end devices and servers in IoT environments. This protocol uses the Elliptic Curve Cryptography-based key exchange mechanism and one-way hash function-based message authentication method to achieve mutual authentication, user anonymity, and forward security. The authors performed formal security proof under the standard model with elliptic curve encryption computational assumptions, and an automatic formal verification was performed with ProVerif. The proposed scheme reduced computation and communication costs while providing improved security features, addressing the challenges posed by IoT deployment.
The paper is well-organised and written. The literature review is apropriated. The content of this article is very significant because we and our smart devices are vulnerable to many cyberattacks. Novel methods that improve communication security in IoT are highly desirable. Thanks to we can feel more secure. Two remarks:
- the authors should correct some typos
- the authors should expand their conclusions section because it is poor
I suggest publishing this article after correcting the mentioned remarks.
Author Response

(The authors gave the same response as above.)

Reviewer 3 Report
Comments and Suggestions for Authors
The paper discusses an improvement on the cryptography algorithm of Srinivas et al [23]. It adds additional security features, a formal verification and implementation details. It is well written, and should be accepted for publication. My remarks mainly concern the introductory parts.
First, as the algorithm relies on [23] and makes additional improvements to the algorithm presented there, this should be clear from the first few sentences in the Introduction, and not to be hidden until Section 3.
Second, I suggest to add a glossary of the numerous abbreviations; keep only those which are really necessary. Here is a list of some of them from page 2: IoT, AKA, ECC, PUF, BR, CK, CRP, WBAN, KCI, PFS, IM, WSN, ESL, MIM, SG, KCI, TA. By the way, what are WSN, MIM, IM?
Third, in cryptography "correctness in standard model" implies that properties of hash functions are derived from other complexity assumptions (and they are not modeled as random functions). As this is not the case here, to avoid confusion I suggest using a different terminology when referring to correctness by the original Canetti-Krawczyk (CK) security model (lines 9, 11, 15, 101, 110).
Line 5: "validated to date, but most" => "used in practice, several of them"
Lines 6-7: "not guaranteed to be secure in real applications" Best to leave out this sentence (or at least the second half of it) as it expresses some half-truth. The "model" cited in line 11 suffers from the same shortage.
Line 8: "To reduce the weaknesses, we present" => "We present"
Line 11: "standard model", see my remarks above.
Line 23: "susceptible *to attacks* and *maintaining message integrity is* critical"
Line 26: "investigated" => "used"
Line 30: "As a result, mutual authentication" -- delete "implementing"
Line 48: delete "Until now"
Line 51: "that this scheme could achieve" => "that their scheme achieves"
Line 62: remove "Regrettably"
Line 64: "covers available problems" => "offers solutions to some of the previously mentioned problems such as"
Lines 65-67: I could not comprehend the sentence starting with "Unfortunately ...". TA is supposed to be trusted, isn't it? Please add reference to this claim. See also the remark to Table 1.
Line 70: "demonstrated *that* scheme"
Line 71: while "showcase" can be used as a verb, I recommend replacing it by some other word.
Lines 79,81: two consecutive sentences starting with "However".
Line 85: "put forward *a* formal" -- this is not the first such a model.
Line 86: the model is not resistant to an attack but attacks can be formalized within the model. Same applies to the next sentence.
Line 91: "remarkably strong" => "somewhat stronger"
Line 98: "weak" => "cannot handle"
Line 101: usage of "standard model", see the initial remarks
Line 108: "reveals security issues" (delete "the")
Line 111: delete "automatically"
Line 120: Section name "INTRODUCTION" is all capital.
Line 134: "actuators" appear here. What are they? What do they do?
Line 138: "p as big as q" => "p > q"
Lines 137-139: Say that the group operation is the usual multiplication of points on the elliptic curve, and G is a subgroup of order p.
Table 1: "Trust Anchor" => "Trusted Authority" (here and elsewhere) T is used for both as a public key and as a timestamp. Use different letters.
Line 147: add the scheme name before the reference number [23]
Figure 4 (above line 182): No need to store PKsp and PKs as they can be computed from the stored Ksp (and ks) values.
Line 430: the same half sentence appears twice.
Author Response
Dear Editor and Reviewers,
We are very grateful for your constructive comments on our submission, applsci-2852529.
We have considered them all when preparing the current revision and this response letter.
I've included for you below the detailed responses to your comments. We cite your different types of comments, followed by indices, and give our answers and actions to the respective comments. We marked the answered content with red and the modified content with blue in the revised manuscript. In addition, we also put the revision in this letter for the respective comments.
We hope the following detailed responses can help the editor and the reviewers better understand our revised manuscript. Thanks for your consideration! We are excited to hear from you.

Reviewer 4 Report
Comments and Suggestions for Authors
The aim of this interesting paper was to point out that endpoint devices in the IoT are often deployed unattended and connected to open networks, which can make them vulnerable to physical tampering and other security attacks. Various authentication key agreement schemes have been validated so far, but most schemes do not cover the necessary security features or are incompatible with resource-constrained end-devices. Furthermore, their security proofs have been done under the "real or random" model, which is not guaranteed to be secure in real applications.The proposal leverages the ECC-based key exchange mechanism and one-way hash function-based message authentication method to achieve mutual authentication, user anonymity, and forward security.
It was made Descriptive Security analysis – anonymity, mutual authentication, ESL attack resistance, impersonation attacks resistance, IoT nodes capture attack resistance, KCI attack resistance.
It is widely recognized that cryptographic schemes proven secure in the RoM model may not necessarily provide the same level of security when implemented in real-world systems. Furthermore, was proposed a security-enhanced AKA protocol for IoT devices to connect to servers. The security of the proposed scheme is rigorously proved under the eCK model with the elliptic curve encryption computational assumptions, and ProVerif verifies the session key confidentiality and authentication properties.
Author Response

(The authors gave the same response as above.)

Reviewer 5 Report
Comments and Suggestions for Authors
Dear Authors,
I am pleased to review your manuscript on "Provably secure ECC-based anonymous authentication and key Agreement for IoT". This topic is very important in the field of embedded systems and IoT.
Overall, the paper is well written is the sense of description of the network model and the further details regarding security. There is however a concern regarding the following issues:
1. As a reader of this topic I find difficult to follow the design of manuscript. It has too many chapters. Perhaps, some reorganization of the content would add more clarity.
2, Table 4 has a missing information on column 4. Please add some units to the presented information. The reader needs to understand the communication cost. Please check and as the specific unit/s to describe the numeric data.
3. A phrase for description of E1-E6 will add clarity to the manuscript. Please introduce some description in order to grasp the meaning of this notation and in chapter 5.
4. Due to the multiple abbreviated terms is difficult to follow the text. I keep searching the meaning of the terms. Please add a keywords table at the end of the manuscript to identify the complete description of these words.
5. Table 7 is not in the right place. Please move table 7 from chapter 9 in chapter 8 where is referenced and add a phase in the manuscript to understand this table, and after that add the conclusion from lines 470-473.
6. Chapter 9, Conclusion section must be redone in order to bring highlights of the manuscript and conclude with more impact.
7. As a reader, some visualization of an attacker in the network would bring more visibility to the manuscript. Please think of that and add it where is needed.
8. Some limitation to your study/research would bring more impact. It will help other researchers to direct them to your research and add visibility gain.
That's all. Best of luck with your research!
Author Response
Dear Editor and Reviewers,
We are very grateful for your constructive comments on our submission, applsci-2852529.
We have considered them all when preparing the current revision and this response letter. Please find below-detailed responses to your comments. We cite your different types of comments, followed by indices, and give our answers and actions to the respective comments. We marked the answered content with red and the modified content with blue in the revised manuscript. In addition, we also put the revision in this letter for the respective comments.
We hope the following detailed responses can help the editor and the reviewers better understand our revised manuscript. Thanks for your consideration! We are excited to hear from you.

Reviewer 6 Report
Comments and Suggestions for Authors
This paper represents a novel contribution to the literature, but needs to undergo one round of edits to clarify the goals of the paper into polish English language writing.
First, the authors need to rephrase the aims of the study into research questions on page 5. Instead of stating the contributions to the literature, that the research makes, explain to the reader, what research questions you will answer. Then, later in the paper, you can explain what contributions your finished research has made.
Maintain consistent font size throughout the manuscript in narrative sections. There are paragraphs where the font size is too small.
In your conclusion, use past tense to explain the research that you have conducted, and make sure that you remind your reader of the contribution that you made to the literature. If you include research questions earlier in your paper, you will be able to answer those research questions in your conclusion, which will be much simpler for readers to understand.
Comments on the Quality of English Language
Minor editing of English grammar necessary.
Author Response

(The authors gave the same response as above.)

Round 2
Reviewer 3 Report
Comments and Suggestions for Authors
None of my remarks has been addressed. Please do so.
Author Response
Dear Editor and Reviewers,
We are very grateful for your constructive comments on our submission, applsci-2852529. We have considered them all when preparing the current revision and this response letter. We've included below the detailed responses to your comments for you. We cite your different types of comments, followed by indices, and give our answers and actions to the respective comments. We mark the answered content with \textcolor{red}{red} color and the modified content with \textcolor{blue}{blue} color in the revised manuscript. Moreover, we also put the revision in this letter for the respective comments.
We hope the following detailed responses can help the editor and the reviewers better understand our revised manuscript. Thanks for your consideration! We are excited to hear from you.

Round 3
Reviewer 3 Report
Comments and Suggestions for Authors
All my remarks have been addressed.